# Threshold for Relationship between Vitamin D and Parathyroid Hormone in Chinese Women of Childbearing Age

**DOI:** 10.3390/ijerph182413060

**Published:** 2021-12-10

**Authors:** Yichun Hu, Siran Li, Jun Wang, Deqiang Zheng, Huidi Zhang, Wei Yu, Lijia Zhu, Zhen Liu, Xiaoguang Yang, Lichen Yang

**Affiliations:** 1Key Laboratory of Trace Element Nutrition of National Health Committee, National Institute for Nutrition and Health, China CDC, Beijing 100050, China; huyc@ninh.chinacdc.cn (Y.H.); sirancara@163.com (S.L.); zhanghuidi1114@126.com (H.Z.); liuzhen@ninh.chinacdc.cn (Z.L.); xgyangcdc@vip.sina.com (X.Y.); 2Physical and Chemical Laboratory, Shenzhen Center for Chronic Disease Control, Shenzhen 518020, China; junwangwh@hotmail.com (J.W.); 550yuwei@163.com (W.Y.); zhu_lijia@foxmail.com (L.Z.); 3Department of Epidemiology and Biostatistics, School of Public Health, Capital Medical University, Beijing 100069, China; dqzheng@ccmu.edu.cn

**Keywords:** 25-hydroxyvitamin D, parathyroid hormone, threshold, Chinese childbearing women

## Abstract

*Background*: The aim of this study was to assess the relationship between serum 25-hydroxyvitamin D [25(OH)D] and serum intact parathyroid hormone (PTH) in Chinese childbearing women, and to estimate the optimum threshold of 25(OH)D that maximally inhibits the PTH, which is considered to be the optimal status for vitamin D sufficiency. *Methods*: Serum samples were selected from the biological samples’ bank of the Chinese Chronic Diseases and Nutrition Survey (CCDNS) 2015. The serum 25(OH)D concentration was determined by liquid chromatography tandem mass spectrometry and the serum PTH was determined by electronic chemiluminescence. Simple linear and partial correlation analysis, locally weighted regression smooth scatterplot (LOESS), nonlinear least squares estimation (NLS), and segmented regression (SR) were utilized to estimate the relationship of 25(OH)D and PTH, and to determine the threshold of 25(OH)D. Results: A total of 1568 serum samples of 25(OH)D concentration and PTH concentration were analyzed. A significant inverse relationship between 25(OH)D and PTH concentration was observed below 15.25 (14.22–16.28) ng/mL, and PTH decreased slowly with the increase of 25(OH)D above 16.75 (15.43–18.06) ng/mL after adjusting by age, latitude, city type, season, corrected calcium, and phosphorus. A very short plateau of PTH was found at 15.25 ng/mL and 16.75 ng/mL in terms of 25(OH)D according to LOESS, NLS, and SR. *Conclusions*: The serum 25(OH)D was negatively correlated with the serum PTH. The threshold of VitD sufficiency was found in the range of 14.22–18.06 ng/mL in terms of serum 25(OH)D concentration for Chinese childbearing women aged 18–44 years old.

## 1. Introduction

Vitamin D (VitD) is one of the essential micronutrients in the human body. VitD can stimulate the synthesis of osteopontin and alkaline phosphatase in osteoblasts and inhibit the apoptosis of osteoblasts, which is beneficial to bone formation [1]. Recent studies have found that VitD nutritional status is not only related to bone health, but also to cell metabolism, immune system, respiratory system, and other human functions [2].

25-hydroxyvitamin D [25(OH)D] is the main circulation form of VitD in blood with a longer half-life of 2–3 weeks than that of 1,25(OH)_2_D and 24,25(OH)_2_D [3,4]. It is also recognized as a reliable indicator for evaluating VitD status in the human body [5]. However, there are still controversies about the threshold of 25(OH)D in serum for judging the nutritional status of VitD. Clinical thresholds to define VitD deficiency and insufficiency were usually based on the estimation that there is a threshold for serum 25(OH)D, and secondary hyperparathyroidism (and bone loss) occurs below this threshold [6]. There are two main popular perspectives of thresholds for VitD sufficiency recommended by the American Institute of Medicine (IOM) and the Endocrine Society (TES), 20 ng/mL and 30 ng/mL as the threshold for VitD sufficient respectively [4,7]. No generally accepted threshold was acknowledged presently.

Intact Parathyroid hormone (PTH) is a calcium-regulated hormone, and it acts mainly on kidney and bone tissues, and even throughout the body [8]. Because of the well-known adverse effects of elevated PTH levels on the skeleton, a biologically relevant value for skeletal physiology would be the level of 25(OH)D, at which serum PTH level increases or no longer decreases [9]. The 25(OH)D level is inversely correlated with PTH and has been regarded as an index of sufficient VitD nutrition. The lowest VitD level at which PTH enters into a platform has been used to define VitD deficiency/insufficiency [10,11]. There are a number of cross-sectional and randomized controlled trials [12,13] that have reported on the inverse relationship between 25(OH)D concentration and PTH concentration in different countries and ethnicities. There was no consistency in the threshold values of serum 25(OH)D and the reported thresholds ranged from 10 nmol/L (4 ng/mL) to 125 nmol/L (50 ng/mL), but the most reported was during 37.5–75 nmol/L (15–30 ng/mL) [6]. There are still some studies in which such thresholds or the negative correlation was not found. The wide range of these thresholds reported may be relevant to the varied ethnicities and ages in the studied populations, the illness which may affect PTH concentrations, varied calcium intake or level in the body, and non-standardization of assays for 25(OH)D concentration [6]. Several studies have shown that PTH increases with age [13], is modified by calcium intake [6,14] and level of phosphate [15,16], or the presence of other co-morbidities [8].

The threshold is important for determining the degree of vitamin D status, especially when it is low enough to increase secondary hyperparathyroidism [15]. In this study, we analyze the data from the Chinese Chronic Diseases and Nutrition Survey (CCDNS) 2015 to assess the relationship between serum 25(OH)D concentration and serum PTH concentration, and to find out whether there is a threshold considered relevant to the bone health in Chinese childbearing women aged 18–44 y.

## 2. Materials and Methods

### 2.1. Study Population and Sampling

All the samples were selected from the CCDNS 2015. CCDNS 2015 is a cross-sectional survey of the civilian non-institutionalized population of China, conducted by the National Institute of Nutrition and Health and The National Center for Chronic and Non-Communicable Disease Control and Prevention, Chinese Center for Disease Control and Prevention (NINH &NCNCD, China CDC). People with serious physical and mental diseases were excluded from the survey. All participants were asked to give informed consent in writing to participate in the survey and the survey was approved by the Institutional Review Board of China CDC (No.201519-A).

The minimum sample size of this study was calculated by the formula below [17,18].
(1)$$N=\frac{p\times \left(1-p\right)\times u^{2}}{d^{2}}\times deff$$

N, number of samples; *p*, deficiency rate; *u*, confidential level, 1.96; *d*, allowance error, 3%; *deff*, design effect, 1. The reported deficiency rate of Chinese women of childbearing age was 44.1% (<20 ng/mL) [19,20]. The calculated minimum sample size was 1053.

Based on the demographic information, all samples of women of childbearing age during 18–44 y in this study were selected from the biological samples’ bank established from CCDNS 2015. Pregnant women were excluded. All the samples were selected by the method of simple random sampling. Considering the representativeness, numbers of surveillance sites (total of 302 sites) and age distribution. We selected one sample for each age group (18–24 y, 25–29 y, 30–34 y, 35–39 y, ≥40 y) at each surveillance site. In addition, considering the possible lack of sample volume and hemolysis in the actual sampling process, we had a small surplus when picking samples. In general, the sampling scheme meets the requirements of the minimum sample size (1053) of the study.

### 2.2. Sample Detection

Samples of 2 mL of fasting venous blood were collected and centrifuged at 1500× *g* for 15 min, 30 min after the blood was taken. The serum was aliquot and stored in a brown vessel at −20 °C in the laboratory where the investigation area was located. All the serum specimens from all the investigation areas were transported to the biological samples’ bank located in NINH by cold chain. All the blood samples were preserved at −70 °C in a freezer before detection.

The liquid chromatography tandem mass spectrometer (AB Sciex Pte. Ltd., Framingham, MA, USA) was used to analyze serum 25(OH)D concentration including 25(OH)D_2_ and 25(OH)D_3_. The sum of 25(OH)D_2_ and 25(OH)D_3_ is recorded as 25(OH)D. The calibration of the assay was verified by using the National Institute of Standards and Technology of America (NIST) standard reference material SRM 972a. The average bias was 2.64% for 25(OH)D_2_ and 3.13% for 25(OH)D_3_ compared with Nist SRM 972a. The PTH was measured by electronic chemiluminescence immunoassay (Roche e601, F Hoffmann-La Roche Ltd., CH4002 Basel, Switzerland). The coefficient of variation (CV) was 4.65% in the 51.5–53.9 pg/mL range, and 4.25% in the 186–191 pg/mL range. The serum album and phosphorus were measured by an automatic biochemical analyzer (7080, HITACHI, Ltd., Tokyo, Japan). The total calcium was detected by inductively coupled plasma mass spectrometry (ICP-MS, PerkinElmer, NexION 350, Waltham, MA, USA). Commercially available quality control samples (Clincheck Level-2, Munich, Germany; Seronorm, Level-2, Billingstad, Norway) were used every 10 samples. The inter- and intra-assay CV were 1.23% and 2.62%, respectively.

### 2.3. Variables

A national project workgroup was established in the China CDC to develop a unified survey and questionnaires to carry out the investigation by using unified equipment and methods. The basic information of the subjects was collected by questionnaires [21]. The city type was divided into urban and rural areas [22]. Latitude was retrieved from a Baidu map (https://map.baidu.com/, accessed on 28 September 2021) and divided by the climate zone of China [23]: below 23.5° N is tropical, 23.5–31.9° N is subtropical, 32.0–40.49° N is warm tropical. The 40.5–46.49° N (middle temperate zone) and above 46.5° N (cold temperate zone) were merged together due to limited sample size. The season was recorded according to the month of blood taken, spring (March to May), summer (June to August), autumn (September to November), and winter (December to February). The calcium was corrected by an album called corrected calcium. Body weight was measured by a uniform electronic scale, and standing height was measured by a metal column type height meter. Body Mass Index (BMI) was calculated according to body weight and height, and the BMI level was classified as thin, normal, overweight, and obese by the cut-points of 18.5 kg/m^2^, 24 kg/m^2^, and 28 kg/m^2^ [24].

### 2.4. Statistical Analysis

Serum 25(OH)D and PTH concentrations were recorded as P50 (P25-P75) because they were not consistent with the normal distribution according to the normality test, and then they were compared by the Kruskal-Wallis test in different subgroups. The rank-based ANOVA was used for pairwise comparison. The simple linear correlation analysis and partial correlation analysis adjustment by confounders including city type, age, latitude, BMI, calcium corrected by albumin, and season were adopted. The 95% confidential intervals were estimated by the bootstraps method. To adjust the potential confounders for threshold estimation of 25(OH)D, the confounders brought into the subsequent analysis were determined by multiple linear regression. After that, we used the generalized additive model (GAM) to adjust the confounders, and then the optimal model for 25(OH)D adjustment was evaluated and determined by the Akaike information criterion (AIC) [25]. Subsequently, the adjusted 25(OH)D was obtained by the optimal model. We adopted the estimation methods described by Wu et al. [26]; the most popular reported model locally weighted regression smooth scatterplot (LOESS) [27] was first used to describe the relationship between 25(OH)D and PTH, and obtain the potential cut-points of 25(OH)D. The nonlinear least squares estimation (NLS) was then used to determine the exact values of cut-point based on the potential cut-points according to the result of LOESS. Finally, the relationship of 25(OH)D and PTH before and after the cut-points was determined by the segmented regression (SR) [28] in terms of slope (beta coefficients) and 95% confidence interval. Thus, we could obtain the threshold for vitD sufficiency based on the above analysis and the given cut-points. The model fitting and threshold estimation was analyzed by the R version 4.1.1 statistical software. The loess () function and segmented package in R software were used. All the other analysis was performed by SPSS version 23.0 statistical software (IBMCorp., Armonk, NY, USA). Two-tailed *p* < 0.05 was considered statistically significant.

## 3. Results

### 3.1. General Characteristics

The serum 25(OH)D concentration and PTH concentration of 1568 blood samples of Chinese childbearing women aged 18–44 y were included in this study. The samples were from 30 provincial administrative regions and covered 279 survey areas from the Chinese mainland. The median age was 31 (25–38) years old. The medium BMI was 22.66 (20.33–25.08) kg/m^2^. The medium albumin was 52.9 (49.7–56.1) g/L. The corrected calcium was 2.26 (2.14–2.39) mmol/L. The concentration of serum phosphorus was 1.34 (1.19–1.49) mmol/L.

### 3.2. Serum 25(OH)D and PTH Status

The medium PTH concentration was 34.3 (25.6–44.5) pg/mL. Significant differences were found in age, latitude, season, and BMI subgroups in terms of PTH. The lowest PTH concentration was found in tropic areas while the highest was found in obese women. The level of PTH in winter was significantly higher than that in spring. The medium 25(OH)D concentration was 16.6 (11.9–22.4) ng/mL, and there were significant differences in different city type, age, latitude, and season subgroups. The medium serum 25(OH)D concentration of 18–24 y was the lowest in all the age groups. Lower 25(OH)D concentration was found in urban areas. Women from tropic areas have a significantly higher 25(OH)D concentration than those from the other areas, followed by subtropical areas. The medium 25(OH)D concentration was lowest in winter. Women with medium exercise intensity had lower 25(OH)D concentrations. No difference was found in BMI subgroups (Table 1).

### 3.3. Relationship between 25(OH)D and PTH

The concentration of 25(OH)D was significantly related with PTH concentration, city type, age, latitude, season, corrected calcium, and phosphorus (Table 2, *p* < 0.05).

The result of partial correlation analysis showed that the serum 25(OH)D concentration was inversely associated with the serum PTH concentration (rs = −0.168, *p* < 0.0001) adjusted by city type, age, corrected calcium, latitude, BMI, season, and phosphorus. As in many studies on the relationship between 25(OH)D and PTH, the negative correlation coefficients reported ranged from −0.15 to −0.45 [6]. PTH and 25(OH)D showed a significantly negative correlation when the level of 25(OH)D was lower than 20 ng/mL, while there was no significant correlation between PTH and 25(OH)D in the range of 20–30 ng/mL and higher than 30 ng/mL (Table 3).

The confounders, city type, age, latitude, season, corrected calcium, and phosphorus were included to adjust the level of 25(OH)D. The adjusted LOESS scatter plots and the segmented regression plot showing the relationship of serum 25(OH)D and PTH are presented in Figure 1. There was only one cut point before adjusting 25(OH)D with the confounders, while two cut points were estimated after adjusting. The cut point given by LOESS and NLS analysis are quite different before adjustment, but the cut point results of NLS and LOESS are in good agreement after adjustment. The unadjusted estimated threshold of 25(OH)D estimated by LOESS was 21.48 ng/mL, while it was 7.82 by subsequent exact NLS. After adjusting the confounders, the potential cut points were 15.35 ng/mL and 18.58 ng/mL (Figure 1A). And the exact cut points were 15.25 (14.22–16.28) ng/mL and 16.75 (15.43–18.06) ng/mL (Figure 1B). The level of PTH decreased sharply with the increase of 25(OH)D when the concentration of 25(OH)D was less than 15.25 ng/mL with a slope of −1.824 (−3.315, −0.512), and then the level of PTH entered to the plateau stage until it reached 16.75 ng/mL. With the increase of 25(OH)D, the level of PTH decreased slowly. There is a short plateau found during 15.25 and 16.75 with the 95% CI of β value being between −2.100 and 8.950 (Figure 1B).

## 4. Discussion

At present, serum 25(OH)D is well acknowledged as a good indicator of VitD nutritional status, but there is no generally accepted threshold for the determination of VitD sufficiency. IOM considered that a serum 25(OH)D level of at least 20 ng/mL (50 nmol/L), meets the requirements of at least 97.5% of the population [5,7]. However, a TES-issued serum 25(OH)D level of 30 ng/mL (75 nmol/L) was considered sufficient [4,29]. The rate of VitD sufficiency of Chinese childbearing women aged 18–44 y of CCDNS was 34.5% according to IOM’s recommendation, and only 5.5% was sufficient when adopting TES’s recommendation (data is not published). Under these circumstances, the determination of the threshold for 25(OH)D, especially based on China’s own population, is very important for evaluating VitD nutritional status, and is critical for the corresponding public health decision-making in China.

In recent years, researchers have paid much attention to defining the appropriate level of VitD and its relationship with health. As early as 1987, researchers had already begun to study the relationship between VitD and PTH by cross-sectional studies or randomized controlled trials to explore the plateau period of PTH inhibition by the level of VitD during different groups, including the elderly, female, children and adolescents, healthy adults, VitD-deficient people, different ethnicities, etc. [30,31]. Survey data from NHANES 2003–2004 and 2004–2005 showed that the relationship among 25(OH)D, bone mineral density, and PTH in American adults varied according to race/ethnicity. A significant inverse relationship between 25(OH)D and PTH concentration was only observed when 25(OH)D concentration was below 26 ng/mL among blacks, while an inverse relationship was observed above and below a 25(OH)D level of 20 ng/mL in whites and Mexican-Americans [30]. Aloia et al. [6] found that PTH reached a plateau stage when the serum 25(OH)D concentration was between 40–50 nmol/L (16–20 ng/mL) of African American women. Okazaki et al. found that 28 ng/mL was identified as a threshold for VitD necessary to stabilize PTH to an optimal level in the Japanese population [32]. Kang et al. found that the VitD level of 18.0 ng/mL could be the inflection point of the maximal suppression of PTH in Korean children aged 0.2–18 y [33]. Wu et al. reported a range of 29–33 nmol/L (11.6–13.2 ng/mL) rather than 50 nmol/L (20 ng/mL) are required for the optimal threshold for middle-aged women in Australia [25]. In addition to the above studies, there are also studies that failed to find the threshold [12,34]. Different races and genetic backgrounds might partially explain the phenomenon, and it is necessary to study whether similar thresholds are suitable for different ethnicities.

As discussed above, the literature adopted the use of PTH in the definition of optimal VitD status. However, the method is still a matter of debate. Wright considers that the inconsistent sensitivity to PTH in different populations will lead to a wide range of thresholds [6]. There is no recognized normal range of serum PTH concentration. To avoid affecting the judgment of hyperthyroidism and hypovitaemia, hospitals at home and abroad usually establish their own reference values in clinical diagnosis. Minieri et al. reported that the range of serum PTH was 15.2–127.7 pg/mL when the 25(OH)D the concentration was below 30 ng/mL, and it was 26.2–89.2 pg/mL when the 25(OH)D concentration was above 40 ng/mL [35]. The reported normal range of PTH based on healthy Chinese adults is 10.78–101.19 pg/mL after excluding parathyroid diseases, kidney diseases, diabetes, and other related diseases [36,37]. A very few samples (10 cases) in this study were lower than the reported normal range of PTH, and we excluded these samples when estimating the threshold. The other samples in this study are all within the normal range reported. Ding et al. found that the concentration of PTH decreased in summer and increased in spring [38]. The PTH concentration was also different in season subgroups in this study, and it was highest in winter. Age also has an impact on the PTH/25(OH)D relationship, which may be owing to some age-related patho-physiologies such as increasing growth factors and different expression levels of VitD receptors [39]. Arabi et al. [13] found that age but not gender modulates the relationship between PTH and VitD by comparing the elderly and adolescents. In addition, the level of PTH in the elderly was, on average, two folds of those in the adolescents at the same VitD level. In this study, we found the difference between age groups was statistically significant in terms of PTH, however the difference in the median PTH of each age group was not so obvious. In addition to the factors mentioned above, the level of calcium and phosphorus was also taken into account and the calcium was corrected by albumin. It is well known that the level of serum calcium can affect the level of PTH [40,41]. VitD deficiency reduces serum calcium levels, leading to PTH synthesis [11,15]. Aloia et al. also considered that it is necessary to ensure adequate calcium when exploring the relationship between 25(OH)D and PTH [6].

Moreover, there is no consensus on the ideal form of the mathematical relationship between PTH and 25(OH)D, although the overall shape of the scatter plots of 25(OH)D and PTH reported in the literature is very similar. In the literature, some parametric statistical models have been proposed, including polynomial (linear, quadratic, cubic), exponential, piecewise quadratic and fractional polynomials, and nonparametric methods [42]. However, different statistical analysis methods might lead to inconsistent 25(OH)D concentration thresholds obtained when analyzing the relationship between PTH and 25(OH)D concentrations [12,42]. In our study, we adopted the most commonly reported models LOESS to analyze the relationship between 25(OH)D and PTH, and then verified the relationship and thresholds by NLS and SR. With the increase of 25(OH)D concentration, PTH concentration tends to be less fluctuated above 15.25 (14.22–16.28) ng/mL. According to the cut points obtained in this study, there is a very short platform period during 15.25 ng/mL and 16.75 ng/mL. Moreover, a relatively gentle negative correlation was found above 16.75 (15.43–18.06) ng/mL in terms of 25(OH)D concentration. The result of the partial correlation between 25(OH)D and PTH also showed a significant negative relationship was found under 20 ng/mL, while no significant correlation was found above 20 ng/mL in terms of 25(OH)D concentration. This was consistent with the cut point range we obtained. We think the small number of participants (less than 35%) above 20 ng/mL may lead to the small change in the relationship between the 25(OH)D and PTH around the second cut point. In addition, there is a certain overlapping in terms of the confidential intervals for the two cut-points. Therefore, we believe that the real threshold for vitamin D sufficiency should be during 14.22 ng/mL and 18.06 ng/mL, considering the confidential intervals.

In summary, we think the threshold of 25(OH)D for sufficiency was found during 14.22–18.06 ng/mL for VitD sufficiency, which is lower than the current recommendation (20 ng/mL) in China [42]. Yao et al. [43] reported the threshold of 50.80 nmol/L (20.32 ng/mL) in adult of both sexes aged 20–45 y in Shanghai city. By using the spline analysis and SR model, Li et al. [44,45] reported that the serum PTH concentration increased rapidly when 25(OH)D was lower than 18.21 ng/mL among 1436 participants from Beijing, Wuhan, Guangzhou, Shanghai, and Chongqing. Bacon et al. [46] found that PTH reached the maximum inhibition when the concentration of 25(OH)D was 16 ng/mL in 221 Chinese women aged 20–35 y by the exponential decay curves analysis. The population is similar with our study and the threshold is very close to that obtained in our study.

We acknowledge several limitations. First, the dietary sources of VitD and calcium intake were not included in this study. We were also limited by the volume of blood samples, the other bone health indicators, endocrine indicators, genetic factors, physical exercise, and other factors that may influence the relationship between PTH and 25(OH)D were not detected. Secondly, the samples did not cover all the seasons owing to the integrate arrangement of CCDNS 2015. Due to the factors mentioned above, bias may occur in the relationship or threshold between PTH and 25(OH)D, therefore the thresholds obtained in this study should be carefully translated.

Yao et al. reported that [43] doses of up to 2000 IU still failed to correct VitD deficiency in 25% Chinese participants, which might be partially due to the effect of genetic factors. Thus, the influence of genetic factors on 25(OH)D cut points should be observed in future studies. Brock et al. found that participants who exercise more tend to have higher 25(OH)D levels [47]. However, since there is no distinction between outdoor and indoor sports, the result may also be affected by sunlight intensity. Lombardi et al. reported that exercise can affect the expression and secretion of PTH by changing the levels of calcium and phosphorus in the circulation [48]. The indicators mentioned above and the use of different health outcomes or indicators which may also affect the relationship and cut point between PTH and 25(OH)D may be incorporated into our future research plan to further determine the exact threshold of 25(OH)D for VitD sufficiency.

## 5. Conclusions

In conclusion, although the present study was hindered by some limitations, we still found a negative relationship between serum 25(OH)D and serum PTH, and the threshold should be at 14.22 ng/mL and 18.06 ng/mL in terms of 25(OH)D concentration in Chinese women aged 18–44 y, which is lower than the current recommendation [42]. Above the threshold, the VitD status might be considered sufficient.

## Figures and Tables

**Figure 1 ijerph-18-13060-f001:**
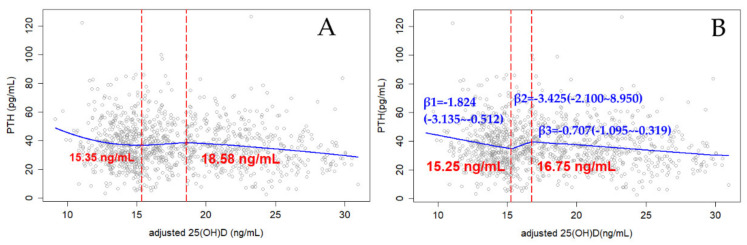
Adjusted relationship between 25-hydroxyvitamin D [25(OH)D] and intact parathyroid hormone (PTH): y-PTH, x-adjusted 25(OH)D; (**A**) locally Weighted Regression Smooth Scatterplot (LOESS); (**B**) Segmented Regression Plot based on the cut-points obtained from the nonlinear least squares estimation; β-Slope of each line segment.

**Table 1 ijerph-18-13060-t001:** 25 hydroxyvitamin D [25(OH)D] and intact parathyroid hormone concentration (PTH) of Chinese childbearing women aged 18–44 y.

Characteristic	N (%)	25(OH)D (ng/mL) *	*p* Value	PTH (pg/mL) *	*p* Value
Total	1568	16.6 (11.9–22.4)		34.3 (25.6–44.5)	
City type			<0.0001		0.912
Urban	663 (42.3)	15.2 (11.2–20.7)		34.2 (25.7–43.9)	
Rural	905 (57.7)	17.8 (12.6–23.9)		34.4 (25.5–44.9)	
Age group, y			0.001		0.044
18~24	372 (23.7)	15.3 (11.3–20.7) ^a^		32.5 (24.1–43.8)	
25~29	363 (23.2)	16.7 (12.1–23.0)		34.0 (25.3–45.1)	
30~34	256 (16.3)	17.0 (12.0–23.4) ^b^		34.0 (23.9–43.8)	
35~39	245 (15.6)	16.7 (12.3–21.8)		36.2 (27.3–44.4)	
≥40	332 (21.2)	17.5 (12.4–23.7) ^b^		34.5 (28.1–45.4)	
Latitude, °N			<0.0001		<0.0001
<23.5	131 (8.4)	25.3 (21.1–29.7) ^a^		27.0 (22.4–35.3) ^a^	
23.5~31.9	628 (40.1)	20.2 (15.5–24.6) ^b^		35.3 (27.1–46.7)	
32~40.4	550 (35.1)	12.8 (10.3–17.0) ^c^		34.1 (25.4–45.0)	
≥40.5	259 (16.5)	13.2 (10.2–17.6) ^c^		34.9 (26.3–44.7)	
Season			<0.001		0.003
Spring	143 (9.1)	16.0 (11.6–21.6)		30.3 (23.0–44.0) ^b^	
Autumn	785 (50.1)	17.4 (12.6–23.0) ^a^		33.1 (25.4–43.2)	
Winter	640 (40.8)	15.6 (11.2–21.8) ^b^		36.0 (26.9–46.4) ^a^	
BMI			0.819		<0.001
Thin	130 (8.3)	15.9 (11.9–21.8)		31.8 (24.7–44.3)	
Normal	895 (57.1)	16.7 (11.9–22.4)		33.0 (24.6–43.5) ^a^	
Overweight	385 (24.5)	16.5 (12.0–23.0)		36.4 (27.7–47.2) ^b^	
Obesity	158 (10.1)	17.2 (12.0–21.5)		37.2 (27.8–44.3)	

^a,b,c^ There were significant differences among groups (*p* < 0.05); * the concentration of 25(OH)D and PTH was expressed as the median and interquartile, P50 (P25-P75); N, numbers of cases.

**Table 2 ijerph-18-13060-t002:** Linear correlation analysis between 25(OH)D, PTH and related variables.

Variables	25(OH)D, Correlation Coefficient (95% CI)	*p* Value
PTH	−0.161 (−0.210–−0.111)	<0.0001
City type	0.153 (0.107–0.200)	<0.0001
Age	0.093 (0.046–0.141)	0.0002
Latitude	−0.541 (−0.575–−0.503)	<0.0001
Season	−0.066 (−0.114–−0.016)	0.010
BMI	0.027 (−0.021–0.081)	0.283
Corrected calcium	0.071 (0.014–0.125)	0.012
Phosphorus	0.083 (0.025–0.138)	0.003

**Table 3 ijerph-18-13060-t003:** Partial correlation between PTH and 25(OH)D under different 25(OH)D levels.

25(OH)D Concentrtaion	Partial Correlation Coefficient (95% CI)	*p* Value
total	−0.168 (−0.227–−0.109)	<0.0001
<12 ng/mL	−0.182 (−0.300–−0.065)	0.003
12–20 ng/mL	−0.123 (−0.210–−0.036)	0.012
20–30 ng/mL	0.001 (−0.095–0.108)	0.980
≥30 ng/mL	0.072 (−0.156–0.296)	0.575

## Data Availability

The original data of this study cannot be made public before the conclusion of the funding project.

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
