# Peer review of "Threshold for Relationship between Vitamin D and Parathyroid Hormone in Chinese Women of Childbearing Age"

_ijerph, 2021, doi:10.3390/ijerph182413060_

Round 1

Reviewer 1 Report

The manuscript investigates thresholds between 25(OH)D and PTH in a Chinese population.

Several details are missing so that the statistical methods aren't fully explained.

Also, it seems that there are no compelling reason to support thresholds in the data, but no comparison with linear trend appears to have been carreid out.

Specific comments:

lines 75-77: It would be helpful if you provided sufficient details to allow the reader to reproduce the sample size calculation. At present details are provided and referring to a standard textbook is useless.

line 76: It doesn't make sense to talk about a minimum sample size. Usually you aim for the sample size that was determined, not a number large than the calculated sample size.

lines 124-129: It's not clear why the authors initially use non-parametric tests but then proceed to use parametric statistical methods for the same outcomes? Please consider using parametric approaches throughout.

lines 129-130: It's not clear how covariates (don't you mean confounders?) were specifically identified? Please more details.

lines 131-133: Please provide more details on the estimation of the GAM model. It's not entirely clear either what this model is used for.

line 135: Nonlinear least squares estimation requires usually pre-specification of a parametric nonlinear model function. There is no mention of such a function and therefore it's not clear what NLS is in the end used for.

line 145: Please provide a justification for having 1568 samples as only 1053 was needed according to the sample size calculation provided above.

Table 1: Please indicate in a footnote or elsewhere which numbers are shown in the table. Pairwise comparisons with accompanying 95% confidence interval would be very helpful instead the present numbers, which seems to be purely descriptive, part form the p-values.

Table 2: This table isn't acceptable. You can't provide estimates without confidence intervals or standard errors. Please do improve the table.

Same comment for Table 3.

Table 4: There seems to be no argument or justification for why the segmented regression models are an improvement over a simple linear regression model. Please consider providing some compelling arguments. From Figure 1 it's not clear that there are thresholds.

Author Response

Point 1 Also, it seems that there are no compelling reason to support thresholds in the data, but no comparison with linear trend appears to have been carried out.

Response 1 We added a description of the threshold in the discussion part. With the increase of 25(OH)D concentration, PTH concentration tends to be less fluctuated above 15.25 (14.22~16.28) ng/mL. According to the cut-points obtained in this study, there is a very short platform period during 15.25 ng/mL and 16.75 ng/mL. Moreover, a relatively gentle negative correlation was found above 16.75 ((15.43~18.06) ng/mL in terms of 25(OH)D concentration. In addition, there is a certain overlapping in terms of the confidential intervals for the two cut-points. Therefore, we believe that the real cut-point for vitamin D sufficiency should be during 14.22 ng/mL and 18.06 ng/mL, considering the confidential intervals.

Point 2 lines 75-77: It would be helpful if you provided sufficient details to allow the reader to reproduce the sample size calculation. At present details are provided and referring to a standard textbook is useless.

Response 2 We have described the detail of sample calculation in the 2.1. Study population and sampling part.

Point 3 line 76: It doesn't make sense to talk about a minimum sample size. Usually you aim for the sample size that was determined, not a number large than the calculated sample size.

Response 3 We calculate the minimum sample size to determine the number of samples we need to reach at least if this study is meaningful. In the actual sample selection process, we considered factors such as the number of surveillance sites, the representativeness and age distribution. Therefore, at least 5 samples (1 sample for each age-group, 18-24y, 25-29y, 30-34y, 35-39y, ≥ 40y) are selected for each surveillance site (totally 302 sites), so as to make the samples as representative as possible and meet the requirements of minimum sample size.

Point 4 lines 124-129: It's not clear why the authors initially use non-parametric tests but then proceed to use parametric statistical methods for the same outcomes? Please consider using parametric approaches throughout.

Response 4 We use the Kruskal–Wallis test for the comparison between multiple groups and the rank-based ANOVA for pairwise comparison because the concentration of 25(OH)D and PTH inconsistent with the normal distribution. We adopted the simple linear correlation analysis and partial correlation analysis to show the correlation of two continuous variables, 25(OH)D concentration and PTH concentration.

Point 5 lines 129-130: It's not clear how covariates (don't you mean confounders?) were specifically identified? Please more details.

Response 5 The word should be “confounder”. We are sorry for the misuse and we have corrected the word all through the paper. The selection of all the confounders was based on the biological plausibility of the factor that related with both the outcome and the exposure of interest.

Point 6 lines 131-133: Please provide more details on the estimation of the GAM model. It's not entirely clear either what this model is used for.

Response 6 we used the generalized additive model (GAM) to adjust the confounders, and then the optimal model for 25(OH)D adjustment. The 25(OH)D was adjusted by the following model,

GAM (25(OH)D ~ factor(season) + factor(city type) + s(age) + s(latitude) + s(corrected ca) + s(phosphorus))

Subsequently, the adjusted 25(OH)D was used to fit the relationship of 25(OH)D and PTH by LOESS, predict the potential cut-points and calculate the exact cut-points.

Point 7 line 135: Nonlinear least squares estimation requires usually pre-specification of a parametric nonlinear model function. There is no mention of such a function and therefore it's not clear what NLS is in the end used for.

Response 7 In our last manuscript, we cited the reference [1] for the statistical methods without detailed description. According to your kind suggestion, we explained the details of the statistical method in the revised manuscript to make the readers have a clearer understanding.

We use the most commonly reported LOESS method to fit the relationship between 25(OH)D and PTH, and look for potential cut-points. Then the nonlinear least squares estimation was used to determine the exact cut-points based on the cut-points obtained from the LOESS. And finally, the segment regression was used to calculate the slope before and after the cut-point, and the relationship between 25(OH)D and PTH was confirmed according to the confidence intervals.

  1. Wu F. ; Wills K. ; Laslett L. L. ; et al. Cut-points for associations between vitamin D status and multiple musculoskeletal outcomes in middle-aged women. Osteoporosis International, 2017, 28:505–515.

Point 8 line 145: Please provide a justification for having 1568 samples as only 1053 was needed according to the sample size calculation provided above.

Response 8 1053 is the minimum sample size required for this study. As illustrated at response 3, we considered factors such as the number of surveillance sites, the representativeness, age distribution, and the possibility of insufficient sample volume or hemolysis in the actual sampling process. Thus, we have some surplus in each surveillance site when picking samples. 1568 samples with complete basic information, good quality control in terms of detected indicators were finally included. We think the sample size fully meets the requirements of the minimum sample size and has good national representativeness.

Point 9 Table 1: Please indicate in a footnote or elsewhere which numbers are shown in the table. Pairwise comparisons with accompanying 95% confidence interval would be very helpful instead the present numbers, which seems to be purely descriptive, part form the p-values.

Response 9 We have added footnotes to Table 1. The concentration of 25(OH)D and PTH are showed in terms of P50(P25~P75) due to inconsistent with the normal distribution. We use the Kruskal–Wallis test for the comparison between multiple groups, and the rank-based ANOVA for pairwise comparison.

Point 10 Table 2: This table isn't acceptable. You can't provide estimates without confidence intervals or standard errors. Please do improve the table.

Response 10 Thank you for the kindness suggestion. We have added the 95% confidence intervals to the table.

Point 11 Same comment for Table 3.

Response 11 Thank you for the kindness suggestion. We have added the 95% confidence intervals to the table.

Point 12 Table 4: There seems to be no argument or justification for why the segmented regression models are an improvement over a simple linear regression model. Please consider providing some compelling arguments. From Figure 1 it's not clear that there are thresholds.

Response 12 We deleted table 4 and described the comparison in the related text. We compared the cutoff values obtained by LOESS and NLS/Segmented regression in the revised manuscript, and also compared the changes of cutoff values before and after adjusting 25(OH)D. The paper does not involve the comparison of simple linear regression and segmented regression. The relationship between 25(OH)D and PTH and the potential cut-points of 25(OH)D was estimated by LOESS. On this basis, according to the characteristics of LOESS plot, we use NLS to calculate the exact cut-points and segmented regression for further fitting. According to the comment from the other reviewer, we added serum phosphorus as a confounder and re-analyzed the relationship between 25(OH)D and PTH. In Figure 1, we showed the fitting graph of LOESS and segmented regression, the cut-points and the slope of segmented regression. Inspired by your suggestions on confidence intervals mentioned above, we think the cut-points for vitamin D sufficiency should be during 14.22~18.06 ng/mL, and we discussed it in the discussion section.

Reviewer 2 Report

Inverse relationships between 25OHD and PTH have been studied before. However, no clear consensus for defining vitamin D deficiency based on these studies has been reached. In this study, the author has studied demographic analysis based on age, latitude, city type, season to decipher the threshold of VitD sufficiency. The only concern is that phosphorus and its contribution to VitD deficiency/sufficiency has been neglected.

Minor comments:

  1. Page 2, line 60, the threshold mentioned here is too far to be accepted as there is just one reference. So, I would recommend using the most accepted values 37.5–75 nmol/l (15–30 ng/ml).
  2. Page 2, Line 65, low calcium intake also states that PTH-induced phosphaturia causes a decrease in serum phosphate. Please include this.
  3. Why phosphate levels have been not evaluated in this study? They are very important along with calcium for bone health.

Author Response

Point 1 Inverse relationships between 25OHD and PTH have been studied before. However, no clear consensus for defining vitamin D deficiency based on these studies has been reached. In this study, the author has studied demographic analysis based on age, latitude, city type, season to decipher the threshold of VitD sufficiency. The only concern is that phosphorus and its contribution to VitD deficiency/sufficiency has been neglected.

Response 1 Thank you for your suggestion. We added serum phosphorus as a confounding factor and re-analyzed the relationship between 25(OH)D and PTH. The results showed that phosphorus had significant correlation with 25(OH)D and PTH, and also had a certain influence on the value of cut-points.

Minor comments:

Point 2: Page 2, line 60, the threshold mentioned here is too far to be accepted as there is just one reference. So, I would recommend using the most accepted values 37.5–75 nmol/l (15–30 ng/ml).

Response 2 We have revised the sentence according to your suggestion.

Point 3 Page 2, Line 65, low calcium intake also states that PTH-induced phosphaturia causes a decrease in serum phosphate. Please include this.

Response3 According to your suggestion, we read the relevant literature and added the description of the impact of phosphorus to the PTH in the 3rd paragraph.

Point 3 Why phosphate levels have been not evaluated in this study? They are very important along with calcium for bone health.

Response 3 Thank your again for the suggestion! We tested and supplemented the serum phosphorus concentration to our database.

Reviewer 3 Report

In the present paper, Yichun Hu and coworkers investigated the relationship between serum 25-hydroxyvitamin D [25(OH)D] and serum intact parathyroid hormone (PTH) in Chinese childbearing women, and to estimate the optimum threshold of 25(OH)D that maximally inhibit the PTH which is considered to be the optimal status for vitamin D sufficiency. The authors showed that the serum 25(OH)D was negatively correlated with the serum PTH. Specifically, the threshold of VitD sufficiency was found in the range of 14.73~18.94 ng/mL in terms of serum 25(OH)D concentration for Chinese childbearing women aged 18-44y.

Overall, I think that the present paper is timely, and it could be of interest to the readers of “International Journal of Environmental Research and Public Health” and researchers, in general.

However, I would like to make some suggestions on how to make the paper stronger.

A lack of bone health indicators, endocrine indicators, genetic factors and other factors that may influence the relationship between PTH and 25(OH)D is very important in this context. Please better discuss this crucial limit of manuscript, also considering the current literature.

Please better discuss the possible clinical applications of these data, considering different target organs and related diseases. As a matter of fact, many studies are now showing an association between vitamin D deficiency and cancer, cardiovascular disease, diabetes, autoimmune diseases, and depression.

Diet has a crucial role on Vit D levels. In fact, the fortification of milk with vitamin D in the 1930s was effective in eradicating rickets in the world. Then, the authors, if possible, should consider the dietary pattern of the patients included in the present study; in this way, I feel that the readers can better understand the results obtained and their possible application to clinical practice.

Research has clearly shown identified a positive and direct relationship between exercise and vitamin D levels in the blood, which may provide evidence that exercise may boost vitamin D stores. Please discuss this intriguing topic in your paper.

There is increasing evidence about the role of microbiome in states of health and disease.  Specifically, vitamin D deficiency may contribute to autoimmunity via its effects on the intestinal barrier function, microbiome composition, and/or direct effects on immune responses. So, does the authors plan to assess/consider microbioma composition in their patients? Please make a comment in the discussion section of revised manuscript on a hot topic of current research.

Author Response

Point 1 A lack of bone health indicators, endocrine indicators, genetic factors and other factors that may influence the relationship between PTH and 25(OH)D is very important in this context. Please better discuss this crucial limit of manuscript, also considering the current literature.

Response 1 We have revised this part in the revised version. We selected potential confounders based on the biological plausibility of the factor that related with both the outcome and the exposure of interest.

Point 2 Please better discuss the possible clinical applications of these data, considering different target organs and related diseases. As a matter of fact, many studies are now showing an association between vitamin D deficiency and cancer, cardiovascular disease, diabetes, autoimmune diseases, and depression.

Response 2 Thank you for your suggestion! The focus of this study is to explore the cut-points or threshold of 25(OH)D in the general population, and the cut-points will be used to evaluate the nutritional status of vitD in the population. All the participants were civilian non-institutionalized population of China selected from the Chinese Chronic Diseases and Nutrition Survey (CCDNS) 2015. We have noticed many reports about the association between vitamin D deficiency and cancer, cardiovascular disease, diabetes, autoimmune diseases and depression. This is why we discuss the threshold, because there is no recognized threshold for vitD status at present. It is the first step of our research to explore the threshold/cut-points to determine whether vitD is sufficient or insufficient in the population, and the research on its threshold or requirement under different disease states needs further exploration.

Point 3 Diet has a crucial role on Vit D levels. In fact, the fortification of milk with vitamin D in the 1930s was effective in eradicating rickets in the world. Then, the authors, if possible, should consider the dietary pattern of the patients included in the present study; in this way, I feel that the readers can better understand the results obtained and their possible application to clinical practice.

Response 3 There are two important issues for the vitD nutritional status of people. One is to judge the vitD nutritional status of human body, and the other is to recommend a reasonable dietary structure. We admit that the intake of vitD is very important to the nutritional status of vitD and we can better guide a reasonable diet based on the dietary intake and vitD status in population. However, the focus of this paper is to obtain the cut-points that can determine the vitD nutritional status, and then we can provide dietary guidance accordingly. The discussion of diet factor on the cut-points will certainly bring more information. However, since we already know the 25(OH)D level in the body we can directly discuss the cut-off point firstly. In the future, we will also study the effects of dietary intake and outdoor activities on vitamin D levels, so as to better guide diet and behavior for the Chinese population. Moreover, we are indeed planning a randomized controlled trial of vitD supplementation to further explore the threshold of vitD. In addition, vitD has been clinically reported to be related to exo-skeleton health, such as cardiovascular disease, diabetes, autoimmune diseases, etc. Since these diseases are affected by many factors, we believe that targeted research is needed to study the vitamin D requirements of clinical specific diseases.

Point 4 Research has clearly shown identified a positive and direct relationship between exercise and vitamin D levels in the blood, which may provide evidence that exercise may boost vitamin D stores. Please discuss this intriguing topic in your paper.

Response 4 Thank you for your suggestion! Some literature reports that exercise has a certain impact on both serum vitamin D and PTH, which is likely to affect the relationship between them. Unfortunately, in our CCDNS 2015, we did not conduct a good investigate on the level and the intensity of physical exercises. In the CCDNS 2015, only the days of 10 minutes of high or medium intensity exercise per week and the time of daily meditation were recorded. We think it is difficult to show the actual level of physical exercises, so it was not included in this study. In our next RCT, we will monitor the indicators of physical exercise in detail to better study its impact on the cut-off point of 25(OH)D and PTH. Nevertheless, we added the topic in the discussion part.

Point 5 There is increasing evidence about the role of microbiome in states of health and disease.  Specifically, vitamin D deficiency may contribute to autoimmunity via its effects on the intestinal barrier function, microbiome composition, and/or direct effects on immune responses. So, does the authors plan to assess/consider microbioma composition in their patients? Please make a comment in the discussion section of revised manuscript on a hot topic of current research.

Response 5 Thank you for your suggestion! We realize that vitD status will affect the composition of gut flora and further affect autoimmune diseases. However, it is unclear whether the gut microbiome composition will affect the level of 25(OH)D or PTH. This study is more focusing on the factors affecting 25(OH)D or PTH or the relationship between them to explore the cut-points for 25(OH)D. Therefore, it is difficult for us to discuss the impact of gut microbiome composition on the cut-points of 25(OH)D for evaluating the vitD nutritional status.

Round 2

Reviewer 1 Report

Thanks for the additional explanations and for the improved manuscript with more detail. Methods and results are easier to follow now.

Reviewer 3 Report

Thank you for addressing my comments well. I have no further remarks.